# Mechanical and Antimicrobial Polyethylene Composites with CaO Nanoparticles

**DOI:** 10.3390/polym12092132

**Published:** 2020-09-18

**Authors:** Cristián Silva, Felipe Bobillier, Daniel Canales, Francesca Antonella Sepúlveda, Alejandro Cament, Nicolás Amigo, Lina M. Rivas, María T. Ulloa, Pablo Reyes, J. Andrés Ortiz, Tatiana Gómez, Carlos Loyo, Paula A. Zapata

**Affiliations:** 1Grupo Polímeros, Facultad de Química y Biología, Universidad de Santiago de Chile USACH, Casilla 40, Correo 33, Santiago 8320096, Chile; cmfa01@gmail.com (C.S.); felipe.bobillier@usach.cl (F.B.); daniel.canales@usach.cl (D.C.); francesca.sepulveda@usach.cl (F.A.S.); alejandro.cament@usach.cl (A.C.); nicolas.amigo@usach.cl (N.A.); jonathan.ortizn@usach.cl (J.A.O.); carlos.loyo@usach.cl (C.L.); 2Genomics and Resistant Microbes (GeRM) Group, Clínica Alemana, Universidad del Desarrollo, Santiago 8320000, Chile; lrivas734@gmail.com; 3Programa de Microbiología y Micología, ICBM-Facultad de Medicina Universidad de Chile, dirección, Avenida Independencia 1027, Comuna Independencia, Santiago 8320000, Chile; mtulloa@med.uchile.cl; 4Centro de Excelencia en Nanotecnología (CEN) Chile, Av. Mariano Sánchez Fontecilla 310, 701-D, Las Condes, Santiago 8320000, Chile; preyes@fundacionleitat.cl; 5Fundación Leitat Chile, Av. Mariano Sánchez Fontecilla 310, 701-D, Las Condes, Santiago 8320000, Chile; 6Departamento de Ingeniería Química, Biotecnología y Materiales, Facultad de Ciencias Físicas y Matemáticas, Universidad de Chile, Avenida Beaucheff 851, Santiago 8320000, Chile; 7Theoretical and Computational Chemistry Center, Institute of Applied Chemical Sciences, Faculty of Engineering, Universidad Autónoma de Chile, Avenida Pedro de Valdivia 425, Santiago 8320000, Chile; tatiana.gomez@uautonoma.cl

**Keywords:** CaO nanoparticles, nanocomposite, polyethylene matrix, mechanical properties, ion release (Ca^2+^), biocidal activity

## Abstract

Low-density polyethylene composites containing different sizes of calcium oxide (CaO) nanoparticles were obtained by melt mixing. The CaO nanoparticles were synthesized by either the sol-gel or sonication methods, obtaining two different sizes: ca. 55 nm and 25 nm. These nanoparticles were used either as-synthesized or were modified organically on the surface with oleic acid (Mod-CaO), at concentrations of 3, 5, and 10 wt% in the polymer. The Mod-CaO nanoparticles of 25 nm can act as nucleating agents, increasing the polymer’s crystallinity. The Young’s Modulus increased with the Mod-CaO nanoparticles, rendering higher reinforcement effects with an increase as high as 36%. The reduction in *Escherichia coli* bacteria in the nanocomposites increased with the amount of CaO nanoparticles, the size reduction, and the surface modification. The highest antimicrobial behavior was found in the composites with a Mod-CaO of 25 nm, presenting a reduction of 99.99%. This strong antimicrobial effect can be associated with the release of the Ca^2+^ from the composites, as studied for the composite with 10 wt% nanoparticles. The ion release was dependent on the size of the nanoparticles and their surface modification. These findings show that CaO nanoparticles are an excellent alternative as an antimicrobial filler in polymer nanocomposites to be applied for food packaging or medical devices.

## 1. Introduction

Polymeric nanocomposites are defined as mixtures of a polymer matrix with inorganic or organic fillers, having at least one dimension in the nanometer range [1,2]. Compared to neat polymers and micro-particulate-based composites, nanocomposite materials have markedly improved properties, including their elastic modulus, mechanical strength, barrier performance, optical transparency, and solvent and heat resistance, among others [3,4]. These improvements are further achieved at much lower loads of the inorganic component (1–10 wt%) than conventionally filled polymers (25–40 wt%). This behavior is mainly because nanoparticles have larger specific surface areas, generating high interfaces with the polymer [5]. The final properties of the nanocomposites depend on the nature and effectiveness of interactions at the polymer/filler interfacial region, which are tuned by both the surface area and the dispersion of the particles in the matrix.

The incorporation of nanoparticles in a polymer can not only improve its general physical properties, but it can also add new functionalities—for instance, biocidal characteristics [6,7,8,9]. These materials constitute an attractive alternative to be used in the medical or food packaging fields. The mechanisms can be based on either catalytic biocidal activities or the release of some active agents from the nanoparticles. Examples of antimicrobial nanoparticles used as fillers are silver [8]; copper [9]; and, recently, TiO_2_ and ZnO nanoparticles [10,11], which have been incorporated into polyolefins such as polyethylene. From the different polymer matrices, polyolefins such as polyethylene are highlighted due to their broad range of applications in packaging and medicines [12]. However, these nanoparticles have some disadvantages; authors have reported that the silver, copper, and ZnO nanoparticles have shown cytotoxicity and genotoxicity, affecting different types of cells [13], and that TiO_2_ nanoparticles require external stimulus, such as UV irradiation, in order to obtain antimicrobial properties [10]. It is important to use antimicrobial nanoparticles that do not require UV irradiation and are nontoxic, therefore calcium oxide (CaO) nanoparticles could be interesting antimicrobial nanoparticles.

Calcium oxide nanoparticles are stressed, as they are biocompatible and very promising antibacterial agents [14]. CaO nanoparticles can eliminate 99% of pathogens such as *Escherichia coli* when they are exposed to 0.05% of these particles [15]. They presented bactericidal action against *Salmonella typhimurium*, *Staphylococcus aureus*, and *Bacillus subtilis* [15,16,17]. Moreover, unlike TiO_2_ and ZnO nanoparticles, which require UV illumination to cause significant antibacterial activity, CaO nanoparticles did not require any external stimulus such as a light source to show antibacterial and antifungal effects [17]. The antimicrobial activity of calcium oxide is due to the presence of water, which generates the Ca^2+^ cation which is responsible for passing through the bacterial membrane, causing cell death [18].

Another advantage of the biocidal behavior of CaO nanoparticles is that the mechanism is not only related to the generation of reactive oxygen species (ROS) on the surface, as for other metal oxides, but also to the pH increase by hydration, forming hydroxides and releasing Ca^2+^ ions [19,20]. Gedanken et al. [21] studied also the bactericidal effect of CaO nanoparticles on Gram-positive (*Staphylococcus aureus*) and Gram-negative (*Escherichia coli)* bacteria, showing these possible mechanisms.

Sawai et al. [22] showed that the active oxygen species generated on the surface of CaO particles was in the form of superoxide radicals (O_2_^−^); the O_2_^−^ activity may be localized on the cell surface. On the other hand, O_2_^−^ captures hydrogen ions and produces hydroperoxyl radicals (HO_2_^•^); HO_2_^•^ is more reactive than O_2_^−^ and is able to penetrate the cell membrane [23]. Gedda et al. [24] studied the effect of CaO on *E. coli* and *S. aeurus,* attributing the biocidal effect to the generated ROS, interacting with the carbonyl groups that are present in the peptide bonds in the cell wall and promoting protein degradation, which results in the destruction of the cell wall.

Moreover, the penetration of these nanoparticles into the bacterial membrane can also occur due to their high surface-to-volume ratio, further explaining their biocidal behavior. The latter shows that smaller particles are more active in killing the bacteria, confirming the relevance of particle size. Compared to TiO_2_, Ag, and Cu nanoparticles, CaO nanoparticles have several other advantages, such as stability under harsh processing conditions; synthesis from inexpensive precursors; and low-toxicity and high biocompatibility, which mean they generally regarded as safe materials for humans [18,25]. Due to all the characteristics of the CaO nanoparticles, they are of great interest to be incorporated into the polymer matrix and to be used as antimicrobial materials in food packaging or medical devices.

The incorporation of CaO nanoparticles into a polymer matrix has been much less studied than other systems. For instance, poly(ε-caprolactone) (PCL) and PCL/gelatin (1:1 *w/w*) were loaded with ca. 60 nm size CaO nanoparticles (0%, 5%, 10%, and 15%) for the treatment of osteomyelitis and bone tissue engineering. Although consistent, antibacterial activity against *S. aureus* was not observed using these nanoparticles, and the presence of CaO nanoparticles in the polymer enhanced the cell viability [26] length with the addition of a filler [27]. The influence of the antimicrobial properties was not studied.

Low-density polyethylene with nano-CaCO_3_ was used for packaging fresh sugarcane. The transmission rate of O_2_ and CO_2_ of the nano-CaCO_3_-LDPE material was lower than that of neat low-density polyethylene (LDPE). It should be noted that the nano-CaCO_3_-LDPE packing and total bacterial count (TBC) of fresh-cut sugarcane were significantly reduced [28].

More studied is the incorporation of calcium carbonate, which is a precursor of calcium oxide when calcined above 700 °C. There are some studies of the effect of the incorporation of CaCO_3_ in a polyethylene (PE) matrix on the mechanical and rheological properties [29,30,31]. The addition of 10 vol% of nano-sized calcium carbonate increased both the yield stress and Young’s modulus of PE [32].

Studies regarding PE/CaO have been barely reported, in particular for their use as antimicrobial materials. Sängerlaub S et al. reported the use of LDPE films with micrometric particles (CaO, <100 µm) as a desicant. The water vapor permeation coefficient of the films with dispersed CaO was a factor up to 24 times higher than that for pure LDPE films. The tensile stress changed only slightly, while the tensile strain at break was reduced with greater CaO concentrations, from 31.8% (pure LDPE) to 10% (LDPE/CaO) [33]. Torrano H.C et al. reported composites obtained by a melting process based on high-density polyethylene (HDPE) with the CaO microparticles from mollusk shell powder, with a particle size from 0.8 to 36.5 µm. The HDPE/CaO showed a reduction in the melting fluid index of ca. 20% with CaO incorporation compared to that of neat HDPE. The HDPE/CaO had a higher thermal stability than neat HDPE. The dynamic mechanical properties were evaluated; the storage and loss modulus of the HDPE/CaO increased with the number of particles compared to neat PE. The influence on the antimicrobial properties was not studied [34].

In another study, waste eggshells were utilized to produce calcium oxide (E-CaO) (macroporous structure) and hydroxyapatite (E-HAP) (50–100 nm), which were melt-compounded with linear low-density polyethylene (LLDPE). The prepared E-CaO and nanostructured E-HAP/LLDPE polymeric composites showed an improvement in hardness, impact strength, and tensile strength with the addition of the filler [35].

As mentioned in state of the art research, obtaining films based on polymers/CaO nanocomposites with high antimicrobial properties and the influence of CaO size on the mechanical and antimicrobial properties have not been reported.

Therefore, CaO nanoparticles of 55 nm and 25 nm were synthesized by the sol-gel or sonication methods, respectively. The nanoparticle surface was modified with oleic acid (Mod-CaO) in order to improve the interaction between the nanoparticles and LDPE. The CaO and Mod-CaO nanoparticles in different amounts (3, 5, and 10 wt%) were incorporated into the LDPE by the melting process. The effects of the size, modification, and amount of nanoparticles of LDPE/CaO and LDPE/Mod-CaO on the thermal, mechanical, and antimicrobial properties were studied.

## 2. Materials and Methods

### 2.1. Materials

CaO nanoparticles of ca. 25 nm size were prepared by a sonication method using Ca(NO_3_)_2_·4H_2_O Sigma-Aldrich 99%(Darmstadt, Germany), ethylene glycol Merck 99.5% (Kenilworth, NJ, USA), and NaOH pellets 99% Mallinckodt Chemical (Dublin, Ireland).

The reagents used for the organic nanoparticle modifications were oleic acid 90% Sigma-Aldrich (Darmstadt, Germany) and hexane 99.8% as solvent J.T. Baker (Phillipsburg, NJ, USA). The low-density polyethylene (LDPE) pellets with 0.93 g/cm^3^ and a melt flow index (MFI) of 116 °C were used as a matrix and were purchased from the B.O. packaging S.A. (Santiago, RM, Chile).

### 2.2. Synthesis of CaO Nanoparticles

#### 2.2.1. Nanoparticles of CaO ca. 55 nm

CaO nanoparticles were synthesized using the method reported by Mirghiasi [14]. Two solutions were prepared at room temperature. The first solution, Ca(NO_3_)_2_·4H_2_O (1 M) (Solution 1), was stirred mechanically for 5 min at 500 rpm and 30 °C. The second solution, citric acid (2.5 M) (Solution 2), was stirred manually until homogenized. Solution 2 was added dropwise to solution 1, and the mixture was dried at 120 °C overnight until gel formation. The materials were ground to a powder and were calcined completely at 900 °C for 5 h.

#### 2.2.2. Nanoparticles of CaO ca. 25 nm

CaO nanoparticle synthesis (SNp) was reported by Tang et al. [32]. The Ca(NO_3_)_2_·4H_2_O (2M) was dissolved in ethylene glycol (25 mL), and then NaOH (4.2 M) was added to the solution under sonication. The mixture was sonicated for 10 min and then kept for 5 h at ambient temperature. The precipitate was removed by filtration and washed with water. Finally, the CaO nanoparticles were calcined at 500 °C for 5 h.

#### 2.2.3. Organic Modification of CaO Nanoparticles (Mod-CaO)

The nanoparticles were modified with oleic acid using the method reported by Li, Zhu, and our research group [36,37,38,39]. Oleic acid (400 µL) and n-hexane (50 mL) were mixed with stirring. Then, 0.5 g of CaO nanoparticles was added to the solution, which was sonicated for 20 min to improve homogenization. The solution was kept under a nitrogen atmosphere with vigorous stirring for 5 h at 60 °C. The nanoparticles were filtered, washed with ethanol, and vacuum-dried at 90 °C for 24 h.

### 2.3. Preparation of LDPE Nanocomposites with the Incorporation of CaO and Modified CaO Nanoparticles

LDPE/CaO composites were prepared using a Brabender Plasti-Corder internal mixer at 120 °C, with stirring at 110 rpm for 10 min. Predetermined amounts of the CaO nanoparticles and neat polymer (as received) were mixed in a nitrogen atmosphere to obtain nanocomposites with 3, 5, and 10 wt% of CaO nanofiller. The samples were press-molded at 170 °C at a pressure of 50 bar for 3 min and cooled under pressure by flushing the press with cold water.

#### 2.3.1. Characterization of CaO Nanoparticles and the LDPE /CaO Nanocomposite

##### CaO Nanoparticles and their Organic Modification Characterization

The morphology and size of the nanoparticles were studied in a TEM electron microscope at 80 kV Philips (Tecnai 12, Amsterdam, The Netherlands). The images obtained were processed using the ImageJ 1.49q software. The X-ray diffraction patterns of the CaO nanoparticles were analyzed on a Siemens D5000 diffractometer (Billerica, MA, USA) using Ni-filtered Cu Kα radiation (λ = 0.154 nm). Fourier transform infrared (FTIR) analyses of the CaO and Mod-CaO nanoparticles were obtained on a FTIR spectrometer Bruker Optics Ltd. (Bruker Vector 22, Coventry, UK). The IR spectra were collected in the 4000 to 500 cm^−1^ range at room temperature.

##### LDPE /CaO and LDPE /Mod-CaO Nanocomposite Characterization

The distribution of the nanoparticles into the polymer matrix was studied on a Zeiss Ultra 55 Field-Emission Scanning Electron Microscope (FE-SEM) (Zeiss Ultra 55, Jena, Germany). The film nanocomposites were placed on conductive tape and mounted on metal studs, and the samples were coated with gold. Electron microscope images were obtained at 22 °C and 20 kV. The images obtained were processed using the ImageJ 1.49q software.

The melting temperature and enthalpy of the fusion of the neat polymer and nanocomposite samples were measured by Differential Scanning Calorimetry (DSC) (Mettler Toledo Model DSC1/500, Columbus, OH, USA). The samples were heated from 25 to 180 °C at a rate of 10 °C/min, and then cooled to 25 °C at the same rate; the values were taken from the second heating curve to eliminate any thermal history. The percent of crystallinity was calculated using the enthalpy of fusion of an ideal polyethylene with 100% crystallinity ($\Delta H_{f\;}^{0}$: 289 J·g^−1^) [39]. The percent crystallinity of the nanocomposites was obtained from Equation (1), where *x* is the percent of nanoparticles in the polymer [40].
(1)$$X_{c}=\frac{\Delta H_{f}\;\times \;100}{\Delta H_{f\;}^{0}\;\left(1-x\right)}$$

The tensile properties of the polymer and nanocomposites were determined on an HP model D-500 dynamometer. The materials were molded for 3 min in a hydraulic press (HP Industrial Instruments) at a pressure of 50 bar and at 170 °C, and further cooled under pressure with water circulation. Dumbbell-shaped samples with an effective length of 30 mm and width of 5 mm were cut from the compression-molded sheets. The samples were tested at a rate of 50 mm/min at 20 °C. Each set of measurements was repeated at least four times.

The antimicrobial effect of the different samples was determined using the plate count method described by ISO 20143. *Escherichia coli ATCC 25922* was used for the analysis. In brief, the initial number of bacteria present after incubation was calculated by counting the number of colonies in a 10-fold dilution. From a fresh culture, a microbial suspension of 1 × 10^7^ CFU/mL by a densimat bioMérieux^®^ was prepared in a Brain Heart Infusion (BHI) broth plus Triton 100x in a humid chamber. The suspension, 0.5 mL, was placed in contact with every 2.5 cm^2^ sample (neat PE, PE/CaO, and PE/ModCaO). Each sample (control and antibacterial treated) was recovered by suspending in 10 mL of sterile saline solution and then diluted serially to 1/10, 1/100, and 1/1000. Then, 0.2 mL of each dilution was plated in duplicate on trypticase soy agar plates and incubated at 37 °C for 24 h. After incubation, the number of colonies in the Petri dishes was counted and, in this way, the percentage of inhibition of microorganisms in each sample was determined compared to the corresponding control. The percent reduction in the colonies was calculated using the following equation (Equation (2)), which relates the number of colonies from the neat polymer to the number of colonies from the nanocomposites [40].
(2)$$\%Reduction=\frac{CFU_{neat\;polymer}-CFU_{nanocomposite}}{CFU_{neat\;polymer}}\;\times \;100$$

The release of calcium ions (Ca^2+^) from the nanocomposites was assessed by inductively coupled plasma (ICP) spectrometry. The nanocomposite samples, of dimensions 1 × 2 × 0.01 cm, were immersed in 10 mL of deionized water with stirring. The measurements were performed after immersion for 1, 4, 8, 12, 15, 30, and 45 days. For each sample, 3 measurement were carried out. The solutions were then centrifuged at 1800 rpm for 2 min to ensure the absence of any particulate matter in the solution. Each sample were measured three times.

## 3. Results

### 3.1. Characterization of Calcium Oxide Nanoparticles

The nanoparticle morphology and diameter distribution were studied by transmission electron microscopy (TEM), as shown in Figure 1, confirming the synthesis of two nanoparticles with different sizes. Figure 1a,b,e show the nanoparticles synthesized by the sol-gel method with a regular sphere morphology and a diameter of ca. 55 nm. Figure 1c,d,f, on the other hand, show the nanoparticles with a diameter of ca. 25 nm synthesized by the sonication method. The effect of the nanoparticle size on the properties of polyethylene can, therefore, be studied.

The CaO nanoparticles presented the characteristic Bragg reflections in Figure 2 at 34°, 37°, 54°, 64°, and 67°, corresponding to planes (111), (200), (220), (311), and (222), respectively, related to the cubic structure of calcium oxide [24,41,42]. Other peaks at 29°, 47°, and 49°, corresponding to planes (104), (018), and (214), are related to the calcite phase [24], which reveals how the fresh CaO easily reacts with carbon dioxide [43,44]. Furthermore, the presence of a peak at 18°, corresponding to plane (0001), is consistent with the standard structure of portlandite [42], and this is due to the fact that CaO is very hydrophilic and reacts on contact with moisture in the atmosphere to produce small amounts of calcium hydroxide Ca(OH)_2_ [44].

The method used for the organic modification of the CaO nanoparticles was based on Li and Zhu [36] and was verified by the FTIR spectra shown in Figure 3. Our research group had modified other nanoparticles (TiO_2_, zinc, silver, and starch) with oleic acid, showing that the modified nanoparticles present better dispersion in the PE polymer and therefore better properties than unmodified nanoparticles [8,11,37,38]. The characteristic peaks of CaO nanoparticles appeared at 3400 cm^−1^ due to hydrogen-bonded hydroxyl groups (O–H). The band at 1490 cm^−1^ and the weak band at 880 cm^−1^ indicate the C-O bond related to the carbonation of CaO nanoparticles [11]. The spectrum shows that the modified nanoparticles had two bands at 2926 cm^−1^ and 2830 cm^−1^, corresponding to the alkyl chain of oleic acid. The modified nanoparticles had a small signal corresponding to the stretching of the carbonyl group of oleic acid at 1710 cm^−1^, with the decrease in the peak indicating that the carboxylic acid group of oleic acid, –COOH, has reacted with surface hydroxyl groups from the CaO nanoparticles. Three other peaks should appear at 1590 cm^−1^, corresponding to carboxylate groups, and at 1550 cm^−1^ and 1430 cm^−1^, corresponding to COO−, but these peaks overlap with the characteristic peak at 1490 cm^−1^ of CaO [37,38].

Studies have reported the degree of esterification of macromolecules such as pectin and cellulose by infrared spectroscopy. They correlated the intensity peak of the ester carbonyl stretching and the deformation of the C–H bond [45,46]. However, the peak of the carbonyl groups from CaO-modified nanoparticles overlaps with the stretching of the carboxylate groups. The organic modification of CaO nanoparticles with oleic acid was performed by an esterification reaction between hydroxyl and carboxyl groups on the surface of CaO and oleic acid, respectively. Thus, in the FTIR spectra of the organically modified nanoparticles, the intensity of the O-H stretching vibration decreased compared to neat CaO nanoparticles due to the involved esterification reaction. Consequently, it is possible to determine an approximate percentage of oleyl groups as the reduction percentage of the O-H stretching intensity of modified CaO nanoparticles with respect to the O-H stretching of neat CaO nanoparticles [36]. As a result, the oleyl group content is ca. 10.6%.

### 3.2. Characterization of Polymer Nanocomposites

The FE-SEM micrographs of both sizes of the LDPE/CaO and LDPE/Mod-CaO are shown in Figure 4. The pure CaO nanoparticles in the LDPE matrix were well distributed in the whole LDPE matrix, although with zones with some degree of agglomeration reaching the majority of agglomerations in a higher frequency until 1–1.2 µm (histograms, Figure 4f,h). The nanoparticle dispersion was improved when nanoparticles modified with oleic acid (Mod-CaO) were added into the LDPE; the agglomerations decreased, reaching a high frequency until 0.5–0.6 µm (Figure 4g,i), confirming that their modification with oleic acid is an efficient route for improving filler dispersion in the matrix [11]. Therefore, the nanoparticle modification could increase the interaction between the modified nanoparticle surface and the polymer.

#### 3.2.1. Thermal Properties

Figure 5 and Figure 6 displayed the DSC profile for the crystallization temperature and melting temperature of neat LDPE and LDPE/CaO and LDPE/Mod-CaO nanocomposites at 55 and 25 nm, respectively.

Table 1 displays the main results of the thermal characterization of the LDPE and LDPE/CaO and LDPE/Mod-CaO nanocomposite, such as the crystallization temperature, melting temperature, and percent of crystallinity. The incorporation of nanoparticles of ca. 55 nm diameter did not modify the thermal properties of the LDPE, as the Tc and Tm were unchanged. On the other hand, the percent crystallinity increased when Mod-CaO nanoparticles of ca. 25 nm were incorporated into the LDPE, showing that these particles can act as nucleation agents [44].

#### 3.2.2. Mechanical Properties

The mechanical properties of LDPE, LDPE/CaO, and LDPE/Mod-CaO composites under tensile tests are given in Table 2 and Figure 7 and Figure 8. With the incorporation of the CaO nanoparticles, the Young’s Modulus of LDPE/CaO or LDPE/Mod-CaO increased approximately 36.6% compared to LDPE.

The elastic limit for the LDPE/CaO and LDPE/Mod-CaO nanocomposites decreased with the addition of CaO or Mod-CaO nanoparticles with a 55 nm diameter until 55.8% and 33.8%, compared to neat LDPE, respectively. By incorporating CaO or Mod-CaO nanoparticles of ca. 25 nm, the elastic limit of LDPE/CaO increased compared to neat LDPE, and it was independent of the nanoparticle modifications.

The elongation at break decreased with the incorporation of CaO, and it was more evident when CaO of approximately 25 nm was incorporated. The elongation at break for LDPE/Mod-CaO decreased 13% compared to neat LDPE.

#### 3.2.3. Biocidal Properties

The biocidal properties of the LDPE/CaO and LDPE/Mod-CaO nanocomposites against *E. coli* are shown in Table 3. The reduction in *E. coli* bacteria on the nanocomposites increased with the amount of CaO nanoparticles and with the reduction of the CaO diameter and its surface modification.

When the CaO nanoparticles (25 nm) were modified with oleic acid, the bacteria were reduced by 99.99% in the composites with 10 wt% filler, because the homogeneous dispersion of higher amounts of nanoparticles can increase the biocidal effect of the nanocomposite.

#### 3.2.4. Ion Release from LDPE Matrix

Figure 9 shows the results of the ICP analysis of calcium cation released in an aqueous environment for 42 days when 10% of the CaO nanoparticles were incorporated into LDPE. The smaller (25 nm) and modified nanoparticles presented a greater release than the larger nanoparticles. According to the antimicrobial properties, the cation release was dependent on the nanoparticle size and modification. In the first days, the calcium ion release increases, causing the maximum release of calcium ions from the LDPE matrix. In general, after three days the calcium ion release remained constant for 42 days. The initial release of calcium ions is due to particles on the specimen’s surface. Amorphous layers near the surface can accommodate more water molecules, resulting in greater ion release. Similar behavior was obtained from silver and zinc oxide nanoparticles [8,11].

## 4. Discussion

The increase in the percent of crystallinity when Mod-CaO nanoparticles of ca. 25 nm were incorporated into the LDPE could be due to the better dispersion of these nanoparticles, as it has been reported that a poor particle dispersion hinders the effect of the nanofiller on the crystallinity changes [46]. These results are in agreement with similar systems based on titanium nanotubes modified with hexadecyltrimethoxysilane (Mod-TiO_2_) and incorporated into polypropylene, showing that the addition of nanotubes increased the percent of crystallinity and the number of spherulites per unit of area. The presence of the particles reduced also the diameter of spherulites and, in turn, increased the number of spherulites, which means a higher nucleation density. This behavior was more pronounced when Mod-TiO_2_-Ntbs were incorporated into the polymer. The authors explained this behavior, stating that the presence of the nanoparticles distributed in the polymer melt can reduce both the work required to create a new surface and the nucleus size for crystal growth. This behavior occurred because the interface between the polymer crystal and the filler may be less hindered, so the creation of the corresponding free polymer crystal surface increases the nucleation density of the spherulites [32,47].

The improvement in mechanical properties was related to the nanoparticles incorporated and regardless of the surface modifications and sizes of the nanoparticles. These results were unexpected, since some of the Young’s modulus values increased as the size of the particles decreased at the nanoscale [48]. However, some authors stated that the filler size has no effect, or, in some cases, that it has a negative effect on the Young’s modulus due to the poor nanoparticle dispersion and weak filler–polymer interface adhesion [49]. Indeed, Sun et al. explained that when the filler size is reduced to the nanoscale, it is difficult to identify any definite mechanism to explain how size affects the elastic properties of the nanocomposites [50].

The elastic limit of the nanocomposites decreased by incorporating CaO and Mod-CaO nanoparticles compared to neat LDPE. Other authors have reported that the homogeneous dispersion of the nanoparticles in the polymer, increasing the amount of nanocomposite formed, causes a greater in-polymer stiffness. Similar results were found when silica nanoparticles with mean sizes of 12 and 50 nm were incorporated into PA 6 polymer. The SiO_2_ addition to PA 6 leads to increased tensile strength, and smaller particles give better reinforcement [51].

The elongation at break of LDPE nanocomposites shows lower values than that for pure polymer, although with a non-lineal behavior, particularly with Mod-CaO particles of 25 nm diameter. In general, the elongation at break depended on the amount of filler and its dispersion in the polymer matrix, although other variables such as degree of crystallinity can further affect this property [52,53]. The tendency in our results can be associated with a competition between these variables that is different in the composites at 5 and 10 wt%. For instance, Sun et al. concluded that the use of nanometric particles makes difficult an explanation of the correct mechanism for elongation at break [50].

Torrano H.C et al. reported the dynamic mechanical properties at 30–100 °C when micrometric CaO nanoparticles were incorporated into the HDPE. The rigidity of the HDPE/CaO increased with the amount of particles at all temperature ranges in comparison with the neat HDPE. The increase in the storage modulus of the HDPE/CaO was higher at 30 °C [34]. A similar behavior has been found by Kontou et al. [54] for LDPE/SiO_2_ nanocomposites, where the storage and loss modulus increased with the amount of nanoparticles compared to neat LDPE, particularly at low temperatures (−150 to −50 °C). This increase may be due to the fact that macromolecular chains at the interface are restricted by the surface of the filler, greatly limiting molecular motion. An increase in the nanoparticle content enlarges the interfacial area and results in an increased interphase volume.

Regarding the antimicrobial properties, it is known that nanoparticles can travel through the amorphous part of LDPE, so it is probable that nanoparticles smaller than ca. 25 nm travel more easily through the amorphous part of the polymer to reach the surface and form active species. Morones et al. showed that small particle sizes, such as those of silver nanoparticles, facilitate cell penetration, producing a greater biocidal effect [55]. Penetration into the bacterial membrane could be due to their high surface-to-volume ratio, which can also explain the biocidal behavior of these particles, as smaller particles are more active in killing bacteria, which would result in the formation of more active oxygen species per unit weight, confirming the relevance of particle size, which, in turn, is related to the ease with which small individual particles penetrate the membrane of the bacteria. For instance, Leung et al. [56] evaluated the biocidal behavior of MgO nanoparticles, which are similar to CaO nanoparticles, by electron microscopy, ATR-FTIR, and proteomics studies. Changes in the FTIR spectra of the bacteria and lipopolysaccharides (LPS) indicated that the mechanism was based on the interaction between the nanoparticles and the bacterial cell. Membrane damage probably occurs due to a combination of the attachment of the nanoparticles to the membrane and the effects of pH change by ion release. It has been reported that the good dispersion of TiO_2_ nanoparticles within the polymer matrix is a key feature in optimizing the biocidal activity in nanocomposite materials [57].

The antimicrobial results may be related to the release of Ca^2+^ from the LDPE matrix and to the mechanism associated with the increase in pH by the hydration of CaO with water, forming hydroxides and releasing Ca^2+^ ions [18]. Ro et al. state that the biocidal effect may be due to alkalinity, with the hydration of CaO as one of the basic antibacterial mechanisms. Besides this, they found that the bactericidal activity of CaO powder was greater than that of NaOH solution at the same pH, and therefore the antibacterial mechanism of CaO is not only due to alkalinity, but also to the action of activated oxygen generated from CaO [16]. Other authors explained that antimicrobial activity could be associated with the generation of reactive oxygen species (ROS) on the surface, such as the generation of superoxide anion on the surface of these particles [58,59].

The results of these studies provide evidence that CaO has the potential to be used as a powerful antimicrobial agent for manufacturing safe food and medicinal products.

Radheshkumar and Munstedt [59] studied and explained the release process for silver ions from a polymer matrix. This behavior is composed of three elementary processes: water diffusion in the composite specimen; the reaction between the nanoparticles and water molecules, leading to the formation of silver ions; and the migration of silver ions through the composite specimen, leading to their release from the composite to the aqueous environment. The release of silver ions requires their diffusion through the interconnected amorphous portions of the polymer. The release analysis is related to the antimicrobial properties.

## 5. Conclusions

CaO nanoparticles with sizes of ca. 55 nm and 25 nm were obtained by the sol-gel and sonicating methods, respectively, and modified with oleic acid, which was verified by IR analysis. The LDPE/Mod-CaO using ca. 25 nm nanoparticles increased slightly the percentage of crystallinity compared to that of pure LDPE. The increase in Young’s modulus by ca. 35% for LDPE/Mod-CaO was greater than that of pure LDPE, regardless of the nanoparticle size. Nanocomposites containing 10 wt% Mod-CaO killed 99.99% of the bacteria present, showing their excellent efficacy against the bacteria compared to neat PE. The organic surface modification of the nanoparticle may improve the interaction with the polymer matrix and the dispersion. The biocidal properties and cation release increased with the reduction in the nanoparticle size, organic modification, and amount of nanoparticles. Size reduction may improve the diffusion of the nanoparticles through the polymer matrix and the formation of the active species on the surface. Excellent antimicrobial material was obtained for medical applications.

## Figures and Tables

**Figure 1 polymers-12-02132-f001:**
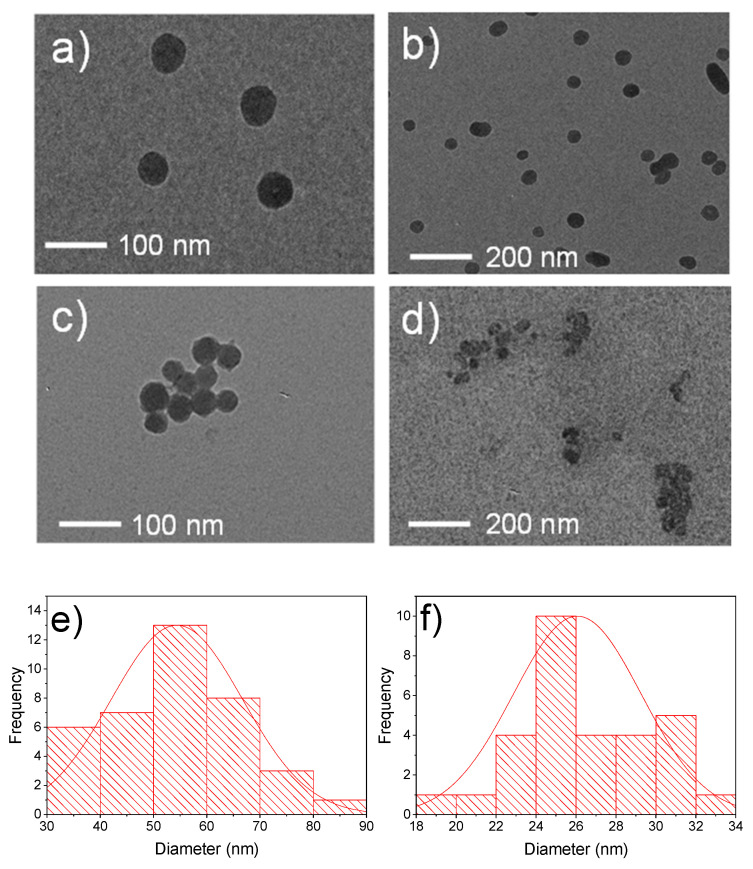
TEM images of the calcium oxide nanoparticles obtained by different routes: (**a**,**b**) CaO nanoparticles ca. 55 nm diameter, (**c**,**d**) CaO nanoparticles ca. 25 nm diameter. Histograms of TEM images of: (**e**) CaO nanoparticles ca. 55 nm diameter and (**f**) CaO nanoparticles ca. 25 nm diameter.

**Figure 2 polymers-12-02132-f002:**
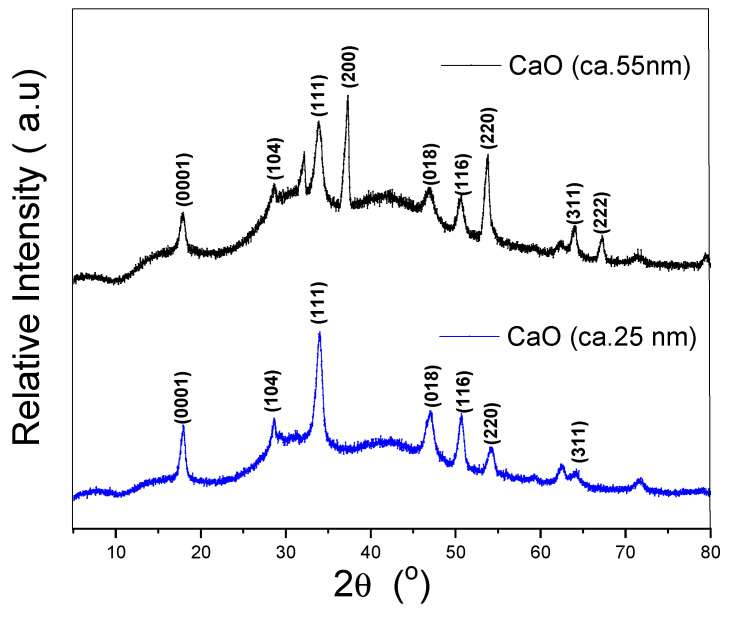
XRD analysis of CaO nanoparticles.

**Figure 3 polymers-12-02132-f003:**
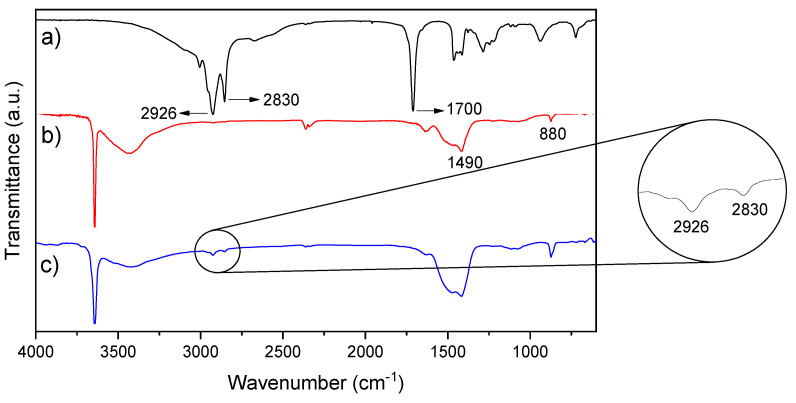
FTIR spectra of (**a**) oleic acid, (**b**) CaO nanoparticles, and (**c**) modified CaO nanoparticles.

**Figure 4 polymers-12-02132-f004:**
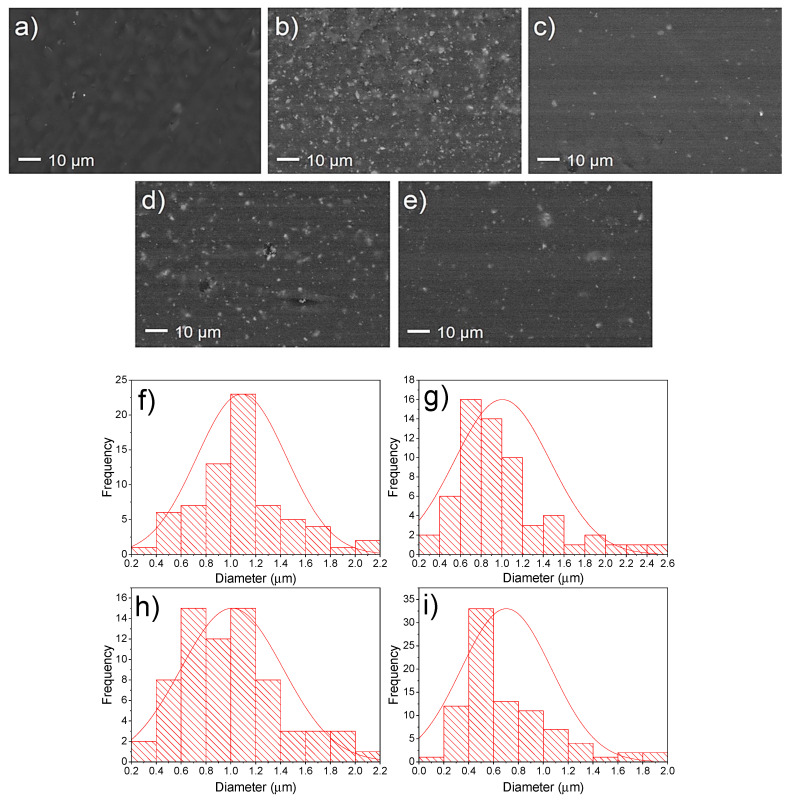
SEM images of (**a**) LDPE, (**b**) LDPE 5% CaO 55 nm, (**c**) LDPE 5% Mod-CaO 55 nm, (**d**) LDPE 5% CaO 25 nm, and (**e**) LDPE 5% Mod-CaO 25 nm. Histograms of SEM images (**f**) LDPE 5% CaO 55 nm, (**g**) LDPE 5% Mod-CaO 55 nm, (**h**) LDPE 5% CaO 25 nm, and (**i**) LDPE 5% Mod-CaO 25 nm.

**Figure 5 polymers-12-02132-f005:**
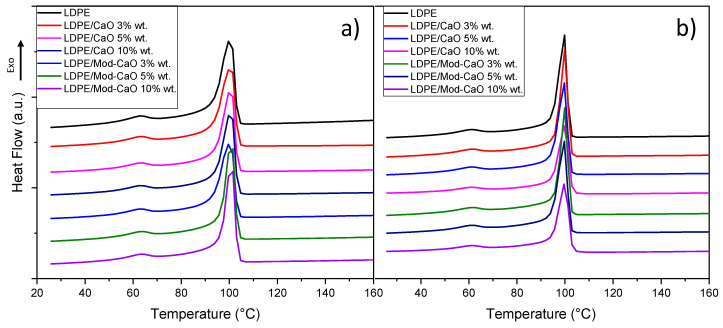
Differential scanning calorimetry of LDPE, LDPE/CaO, and LDPE/Mod-CaO nanocomposites with (**a**) 55 nm- and (**b**) 25 nm-size nanoparticles showing the crystallization temperature (Tc).

**Figure 6 polymers-12-02132-f006:**
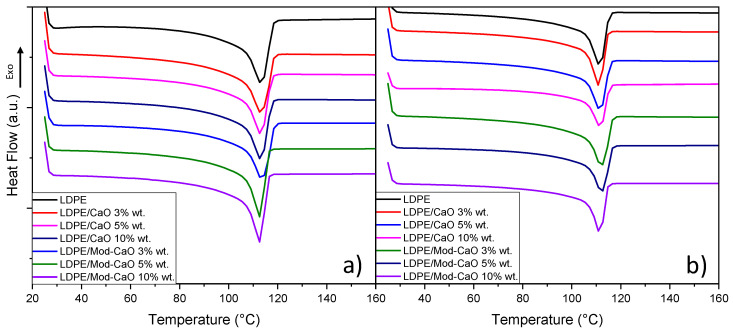
Differential scanning calorimetry of LDPE, LDPE/CaO, and LDPE/Mod-CaO nanocomposites with (**a**) ca. 55 nm- and (**b**) ca. 25 nm-size nanoparticles showing the melting temperature (Tm).

**Figure 7 polymers-12-02132-f007:**
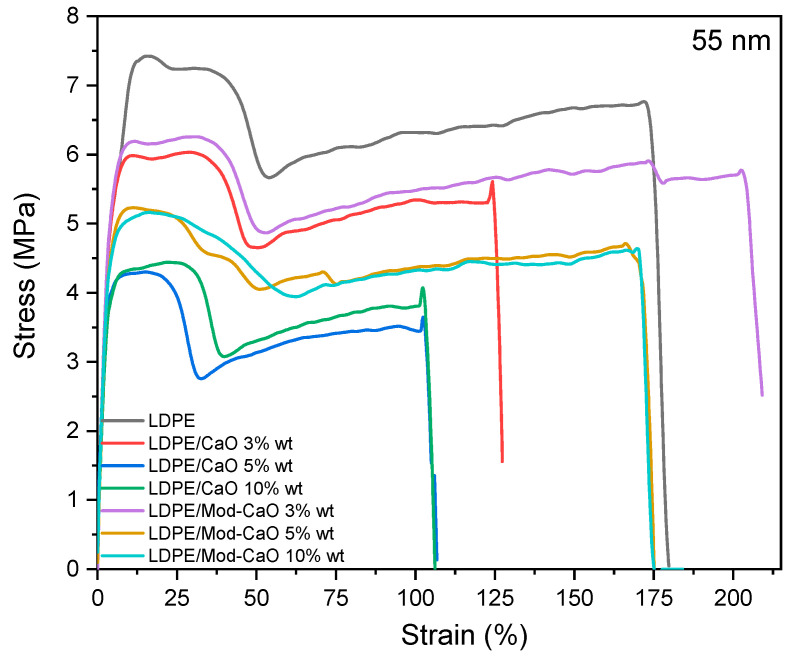
Stress–strain curves of LDPE, LDPE/CaO, and LDPE/Mod-CaO with CaO nanoparticles ca. 55 nm.

**Figure 8 polymers-12-02132-f008:**
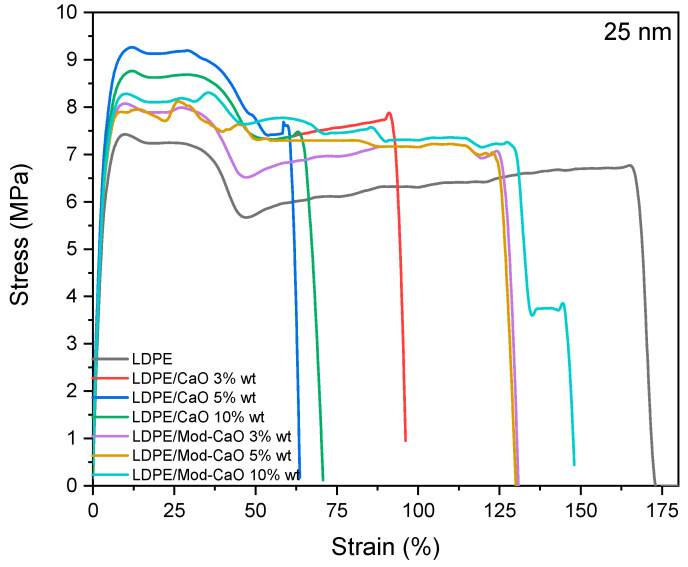
Stress–strain curves of LDPE, LDPE/CaO, and LDPE/Mod-CaO with CaO nanoparticles ca. 25 nm.

**Figure 9 polymers-12-02132-f009:**
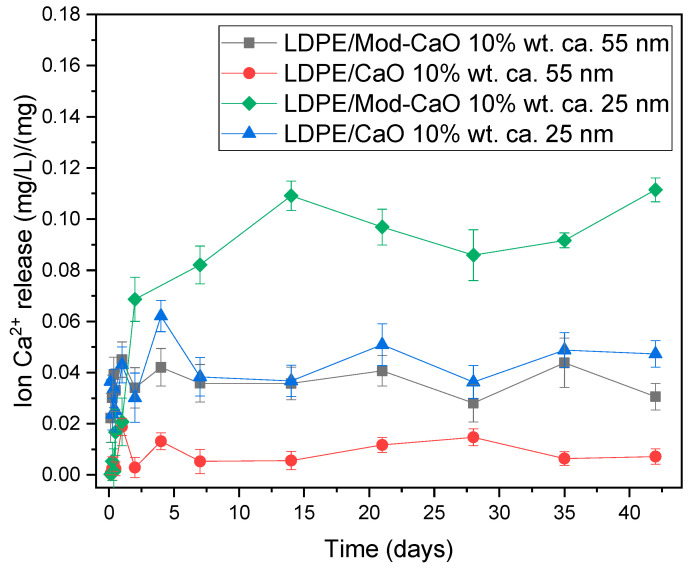
Ca^2+^ release from LDPE/CaO (10 wt% CaO) as a function of time after 42 days.

**Table 1 polymers-12-02132-t001:** Crystallization temperature, melting temperature, and crystallization percentage of LDPE nanocomposites incorporating CaO of approximately 55 nm and 25 nm diameter.

Nanoparticle Size		55 nm			25 nm		
Sample	Weight (%)	T_c_ (°C)	T_m_ (°C)	X_c_ (%)	T_c_ (°C)	T_m_ (°C)	X_c_ (%)
LDPE	0	100	113	40	100	113	40
LDPE/CaO	3	101	113	41	100	111	40
	5	101	112	39	100	112	39
	10	100	113	40	100	112	39
LDPE/Mod-CaO	3	100	113	37	100	113	52
	5	101	112	39	100	113	47
	10	102	113	39	100	113	45

T_c_: Crystallization temperature; T_m_ = melting temperature; X_c_= percent crystallinity; T_10_ = decomposition temperature at 10% weight loss.

**Table 2 polymers-12-02132-t002:** Mechanical properties of CaO/LDPE nanocomposites using nanoparticles of ca. 55 nm and 25 nm diameter.

Nanoparticle Size		55 nm			25 nm		
Sample	CaO Weight (%)	E (Mpa)	Σy (Mpa)	Ee%	E (Mpa)	Σy (Mpa)	Ee %
LDPE	0	150 ± 5	7.7 ± 0.7	169 ± 4	150 ± 5	7.7 ± 0.7	169 ± 4
LDPE/CaO	3	175 ± 16	5.2 ± 0.6	124 ± 4	154 ± 120	8.7 ± 0.6	96 ± 3
	5	165 ± 6	4.4 ± 0.2	104 ± 1	162 ± 24	8.7 ± 0.2	62 ± 2
	10	190 ± 18	4.6 ± 0.4	104 ± 1	193 ± 11	8.8 ± 0.4	70 ± 2
LDPE/Mod-CaO	3	180 ± 12	5.1 ± 0.7	200 ± 14	170 ± 17	8.1 ± 1.1	130 ± 7
	5	195 ± 14	5.4 ± 1.1	164 ± 11	197 ± 12	8.1 ± 0.7	116 ± 2
	10	203 ± 17	5.4 ± 0.6	167 ± 23	205 ± 14	8.6 ± 0.6	147 ± 4

E = Young’s modulus or elastic modulus; σy = elastic limit or yield point; Ee: elongation at break (%).

**Table 3 polymers-12-02132-t003:** Percentage reduction in *E. coli* bacteria for CaO/LDPE nanocomposites using CaO of ca. 55 nm or 25 nm diameter.

Nanoparticle Size		55 nm	25 nm
Sample	Weight (wt%)	% Reduction	% Reduction
LDPE/CaO	3	12.4	61.8
	5	20.2	64.2
	10	30.6	81.4
LDPE/Mod-CaO	3	20.5	80.8
	5	32.2	87.3
	10	53.0	99.9

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
