# Peer review of "Mechanical and Antimicrobial Polyethylene Composites with CaO Nanoparticles"

_polymers, 2020, doi:10.3390/polym12092132_

Round 1
Reviewer 1 Report
In this study, the authors reported the preparation of polymer embedded with CaO nanoparticles with antimicrobial properties. The article is well-written and interesting, thus, it can be reconsidered for publication after a revision process.
1) Low-resolution TEM images should be provided to see whether the morphology and size of CaO nanoparticles are similar. In addition, please supply the size distribution of CaO nanoparticles.
2) Internationally accepted XRD reference should be provided. Also, reflection planes for all peaks should be provided in text or in Figure 2.
3) Only the FTIR data is not enough for characterization, please supply TGA data in addition to that (to see the % of coating).
4) Figure 4. It is highly suggested to perform EDX elemental analysis of the selected area. This can be helpful to track the position and distribution of CaO in a polymer matrix.
5) Please provide the digital pictures of Petri dishes showing the E.coli inhibition with different samples.
Author Response
Chile 9 September 2019
Dear Editor,
We would like to submit the corrections of our paper entitled “Mechanical and antimicrobial polyethylene composites with CaO nanoparticles.” I want to thank the reviewers for their corrections in order to improve the manuscript.
Comments and Suggestions for Authors
Review 1
In this study, the authors reported the preparation of polymer embedded with CaO nanoparticles with antimicrobial properties. The article is well-written and interesting, thus, it can be reconsidered for publication after a revision process.
- Low-resolution TEM images should be provided to see whether the morphology and size of CaO nanoparticles are similar. In addition, please supply the size distribution of CaO nanoparticles.
Answer: The TEM images were improved and the size distribution of the CaO nanoparticles was incorporated in Figure 1.
- Internationally accepted XRD reference should be provided. Also, reflection planes for all peaks should be provided in text or in Figure 2.
This part was improved, the following paragraph was incorporated in lines 236-243:
The CaO nanoparticles presented the characteristic Bragg reflections at 34°, 37°, 54°, 64° and 67° corresponding to planes (111), (200), (220), (311), and (222), respectively, related to the cubic structure of calcium oxide [24, 42,43]. Other peaks at 29°, 47°and 49° corresponding to planes (104), (018), and (214) are related to the calcite phase [24], which reveals how the fresh CaO easily reacts with carbon dioxide [44,45]. Furthermore, the presence of a peak at 18° corresponding to plane (0001) is consistent with the standard structure of portlandite [42], and it is due to the fact that CaO is very hydrophilic and reacts on contact with moisture in the atmosphere to produce small amounts of calcium hydroxide Ca(OH)2 [45].
- Only the FTIR data is not enough for characterization, please supply TGA data in addition to that (to see the % of coating).
You are right, we need to calculate the % of the coating. We do not have to access to TGA at this time, so the following paragraph was incorporated and the % of the coating was calculated from the FTIR spectrum.
The following paragraph was incorporated in lines 258-268:
Studies have reported the degree of esterification of macromolecules such as pectin and cellulose by infrared spectroscopy. They correlated the intensity peak of the ester carbonyl stretching and the deformation of the C-H bond [46,47]. However, the peak of the carbonyl groups from CaO modified nanoparticles is overlapped with the stretching of the carboxylate groups. The organic modification of CaO nanoparticles with oleic acid was performed by an esterification reaction between hydroxyl and carboxyl groups on the surface of CaO and oleic acid, respectively. Thus, in the FTIR spectra of the organically modified nanoparticles, the intensity of the O-H stretching vibration decreased compared to neat CaO nanoparticles, due to the involved esterification reaction. Consequently, it is possible to determine an approximate percentage of oleyl groups as the reduction percentage of the O-H stretching intensity of modified CaO nanoparticles with respect to the O-H stretching of neat CaO nanoparticles[37]. As a result, the oleyl group content is ca. 10.6%.
4) Figure 4. It is highly suggested to perform EDX elemental analysis of the selected area. This can be helpful to track the position and distribution of CaO in a polymer matrix.
Answer: You are right, we improved the resolution and incorporated the histograms in order to study the distribution of the CaO in the LDPE matrix. The following paragraph was incorporated in lines 277-280:
The pure CaO nanoparticles in the LDPE matrix were well distributed in the whole LDPE matrix, although with zones with some degree of agglomeration reaching the majority of agglomerations in higher frequency until 1-1.2 µm (histograms, Figures f and h). The nanoparticle dispersion was improved when nanoparticles modified with oleic acid (Mod-CaO) were added into the LDPE, the agglomerations decreased reaching high frequency until 0.5-0.6 µm (Figures 4g and i) confirming that their modification with oleic acid is an efficient route for improving filler dispersion in the matrix [11].
5) Please provide the digital pictures of Petri dishes showing the E.coli inhibition with different samples.
Answer: We did not take those pictures, but the results were repeated 5 times and the antimicrobial experimental part has been reported on many occasions by our group [8,10,11,38,41,46].

Reviewer 2 Report
This manuscript describes the preparation and antimicrobial property of low-density polyethylene/calcium oxide nanoparticle composites. The composites showed the strong antimicrobial effect, which can be explained to be associated with the release of the Ca+2. As the authors claim, this work indicates that show that CaO nanoparticles are an excellent alternative for antimicrobial filler in polymer nanocomposites. However, due to the following points, I do not recommend publication of this manuscript in current form. Major revisions regarding the follows are required for publication.
- The more previous studies regarding PE/CaO composites should be cited.
- The detailed procedures for the preparation of CaO nanoparticles are described in the sections 2.2.1 and 2.2.2. Therefore, the first two sentences in the section 2.1 are not necessary.
- In the XRD profiles in Figure 2, unidentified peaks are detected, which also should be assigned.
- In the IR spectrum of modified CaO nanoparticles in Figure 3, intensities of the absorption came from oleyl groups are vary low, indicating no occurrence of effective modification. The degrees of functionality should be estimated.
- The SEM images in Figure 4 are unclear for understanding morphologies of the composites. The more precise images with larger magnification should be measured.
- For understanding thermal and mechanical properties more clearly, DSC profiles and stress-strain curves should be provided, respectively.
Author Response
Dear Editor,
We would like to submit the corrections of our paper entitled “Mechanical and antimicrobial polyethylene composites with CaO nanoparticles.” I want to thank the reviewers for their corrections in order to improve the manuscript.
Comments and Suggestions for Authors
Review 2
This manuscript describes the preparation and antimicrobial property of low-density polyethylene/calcium oxide nanoparticle composites. The composites showed the strong antimicrobial effect, which can be explained to be associated with the release of the Ca+2. As the authors claim, this work indicates that show that CaO nanoparticles are an excellent alternative for antimicrobial filler in polymer nanocomposites. However, due to the following points, I do not recommend publication of this manuscript in current form. Major revisions regarding the follows are required for publication.
- The more previous studies regarding PE/CaO composites should be cited.
Answer:
The following paragraph was incorporated in lines 113-124:
The studies regarding PE/CaO have been barely reported, in particular the use as antimicrobial materials. Sängerlaub S et al., reported the use of LDPE films with micrometric particles (CaO, <100 µm) as desicant. The water vapor permeation coefficients of the films with dispersed CaO were a factor up to 24 times higher than for pure LDPE films. The tensile stress changed only slightly, while the tensile strain at break was reduced with greater CaO concentrations, from 31.8% (pure LDPE) to 10% (LDPE/CaO) [34]. Torrano H.C et al., reported composites obtained by a melting process based on high density polyethylene (HDPE) with the CaO microparticles from mollusk shell powder with a particle size form 0.8 µm to 36.5 µm. The HDPE/CaO showed a reduction in the melting fluid index of ca. 20% with CaO incorporation compared to neat HDPE. The HDPE/CaO had a higher thermal stability than neat HDPE. The dynamic-mechanical properties were evaluated, the storage and loss modulus of the HDPE/CaO increased with the amount of particles compared to neat PE. The influence on the antimicrobial properties was not studied [35].
- The detailed procedures for the preparation of CaO nanoparticles are described in the sections 2.2.1 and 2.2.2. Therefore, the first two sentences in the section 2.1 are not necessary.
Answer: You are right. It was corrected, the first two sentences in the section 2.1 were deleted
- In the XRD profiles in Figure 2, unidentified peaks are detected, which also should be assigned.
Answer: You are right. In the Figure 2 all peaks were detected and incorporated into the Figure.
The following paragraph was incorporated in lines 236-243:
The CaO nanoparticles presented the characteristic Bragg reflections at 34°, 37°, 54°, 64° and 67° corresponding to planes (111), (200), (220), (311), and (222), respectively, related to the cubic structure of calcium oxide [24, 42,43]. Other peaks at 29°, 47°and 49° corresponding to planes (104), (018), and (214) are related to the calcite phase [24], which reveals how the fresh CaO easily reacts with carbon dioxide [44,45]. Furthermore, the presence of a peak at 18° corresponding to plane (0001) is consistent with the standard structure of portlandite [42], and it is due to the fact that CaO is very hydrophilic and reacts on contact with moisture in the atmosphere to produce small amounts of calcium hydroxide Ca(OH)2 [45].
- In the IR spectrum of modified CaO nanoparticles in Figure 3, intensities of the absorption came from oleyl groups are vary low, indicating no occurrence of effective modification. The degrees of functionality should be estimated.
You are right, we need to calculated the % of the coating.
The next paragraph was incorporated in lines 258-268:
Studies have reported the degree of esterification of macromolecules such as pectin and cellulose by infrared spectroscopy. They correlated the intensity peak of the ester carbonyl stretching and the deformation of the C-H bond [46,47]. However, the peak of the carbonyl groups from CaO modified nanoparticles is overlapped with the stretching of the carboxylate groups. The organic modification of CaO nanoparticles with oleic acid was performed by an esterification reaction between hydroxyl and carboxyl groups on the surface of CaO and oleic acid, respectively. Thus, in the FTIR spectra of the organically modified nanoparticles, the intensity of the O-H stretching vibration decreased compared to neat CaO nanoparticles, due to the involved esterification reaction. Consequently, it is possible to determine an approximate percentage of oleyl groups as the reduction percentage of the O-H stretching intensity of modified CaO nanoparticles with respect to the O-H stretching of neat CaO nanoparticles[37]. As a result, the oleyl group content is ca. 10.6%.
- The SEM images in Figure 4 are unclear for understanding morphologies of the composites. The more precise images with larger magnification should be measured.
Answer: Figure 4 resolution was improved, and histograms of particles loaded on LDPE were incorporated.
The following paragraph was incorporated in lines 277-280:
The pure CaO nanoparticles in the LDPE matrix were well distributed in the whole LDPE matrix, although with zones with some degree of agglomeration reaching the majority of agglomerations in higher frequency until 1-1.2 µm (histograms, Figures f and h). The nanoparticle dispersion was improved when nanoparticles modified with oleic acid (Mod-CaO) were added into the LDPE, the agglomerations decreased reaching high frequency until 0.5-0.6 µm (Figures 4g and i) confirming that their modification with oleic acid is an efficient route for improving filler dispersion in the matrix [11].
- For understanding thermal and mechanical properties more clearly, DSC profiles and stress-strain curves should be provided, respectively.
Answer: The DSC profiles and the stress-strain curves for all composites were incorporated in the new manuscript
In the Lines 294-296 the following sentence was incorporated.
Figures 5 and 6 displayed the DSC profile for crystallization temperature and melting temperature of neat LDPE and LDPE/CaO and LDPE/Mod-CaO nanocomposites at 55 and 25 nm respectively.
Reviewer 3 Report
Authors describe the thermal, mechanical and antibacterial properties of CaO/PE nanocomposites. The work could be interesting but at the present form there is some lacks that must be addressed before being considered for its publication at Polymers.
Abstract /introduction. The work is adequately summarized but the interest of the work is missing. Why this is work interesting? What is new with respect to the state on art? What is the advancement of this material?
Lines 54 and 55 references are needed.
Lines 70-75- The idea is interesting, could you develop the explanation of ROS alternative mechanism?
The final paragraph of the introduction must be improved, a description of what is carried out in this study must be clear.
Materials.
Plase provide the compreate information of the PE provider.
SEM. Please provide the metal used for the coating.
3.1. The size distribution and the standard deviation must be given. .
Line 194-203. This discussion it is not clear, it is quite mixeitTIrd. Please improve it.
FTIR, the peaks corresponding to the carbonyl groups to the nanocomposite.
3.2. The SEM in the present form do not have much information an additional EDX to evaluate the aggregation of the CaO nanoparticles must be added. A cross section of the nanocomposite also could give an interesting information.
DSC. Add a figure with the thermograms.
Mechanical properties. Provide the diagram of the mechanical properties. I suggest the addition of DMTA analysis to improve the discussion on the effect of the different nanoparticles on the nanocomposite.
Table 2. Please comment why Ee decrases in both cases at 5% but increases again at 10%.
Figure 5. How many measurements have been carried out? Add the error bars for each measure. In addition, the evolution of the thermal and mechanical properties during this experiments should be evaluated.
Discussion, it is difficult to follow the discussion in the present format. The result and discussion must be together.
Author Response
Chile 9 September 2019
Dear Editor,
We would like to submit the corrections of our paper entitled “Mechanical and antimicrobial polyethylene composites with CaO nanoparticles.” I want to thank the reviewers for their corrections in order to improve the manuscript.
Comments and Suggestions for Authors
Review 3
Authors describe the thermal, mechanical and antibacterial properties of CaO/PE nanocomposites. The work could be interesting but at the present form there is some lacks that must be addressed before being considered for its publication at Polymers.
1.Abstract /introduction. The work is adequately summarized but the interest of the work is missing. Why this is work interesting? What is new with respect to the state on art? What is the advancement of this material?.
Answer:
-In the abstract, the following sentence was incorporated:
These findings show that CaO nanoparticles are an excellent alternative as antimicrobial fillers in polymer nanocomposites to be applied for food packaging or medical devices.
-In the introduction, in the lines 61-66, this new sentence was incorporated:
But, these nanoparticles have some disadvantage, authors have reported that the silver, copper and ZnO nanoparticles have shown cytotoxicity and genotoxicity, affecting different types of cells [13] and the TiO2 nanoparticles require the external stimulus as UV irradiation in order to obtain antimicrobial properties [10]. It is important to use antimicrobial nanoparticles that do not require UV irradiation and are nontoxic, therefore calcium oxide (CaO) nanoparticles could be interesting antimicrobial nanoparticles.
-In lines 95-97 the following sentence was incorporated:
Due to all the characteristic of the CaO nanoparticles, they are of great interest to be incorporated into the polymer matrix and to be used as antimicrobial materials in food packaging or medical devices.
-In lines 130-132 the following sentence was incorporated:
As mentioned in the state of the art, obtaining films based on polymers/CaO nanocomposites with high antimicrobial properties and the influence of CaO size in the mechanical and antimicrobial properties has not been reported.
- Lines 58 and 59 references are needed.
The following references [6-9] were incorporated in the new manuscript in line 55:
- https://doi.org/10.1016/j.fpsl.2020.100523,
- https://doi.org/10.1016/j.fpsl.2018.04.001
- https://doi.org/10.1016/j.eurpolymj.2011.05.008.
- https://doi.org/10.1002/marc.200900791.
- Lines 70-75- The idea is interesting, could you develop the explanation of ROS alternative mechanism?
Answer: The new paragraph in lines 82-88 was incorporated:
Sawai et al. [22] showed that the active oxygen species generated on the surface of CaO particles was in the form of superoxide radicals (O2−), the O2− activity may be localized on the cell surface. On the other hand, the O2− captures a hydrogen ion and produces hydroperoxyl radicals (HO2•), the HO2• is more reactive than the O2− and able to penetrate the cell membrane [23]. Gedda et al. [24] studied the effect of CaO on E. coli and S. aeurus, attributing the biocidal effect to the generated ROS, interacting with the carbonyl groups that are present in the peptide bonds in the cell wall, promoting protein degradation which results in the destruction of the cell wall.
- The final paragraph of the introduction must be improved, a description of what is carried out in this study must be clear.
Answer: The final paragraph of the introduction was improved. The paragraph of the new manuscript in lines 133-138 is:
Therefore, CaO nanoparticles of 55 nm and 25 nm were synthesized by the sol-gel or sonication methods, respectively. The nanoparticle surface was modified with oleic acid (Mod-CaO) in order to improve the interaction between nanoparticles and LDPE. The CaO and Mod-CaO nanoparticles in different amounts (3, 5, and 10 wt%) were incorporated into the LDPE by the melting process. The effect of the size, modification and amount of nanoparticles of LDPE/CaO and LDPE/Mod-CaO on the thermal, mechanical and antimicrobial properties were studied.
Materials.
- Please provide the complete information of the PE provider.
Answer: The PE provider of the company is confidential for BO packaging, but BO packaging brings us the density and MFI values. These values were incorporated in the materials section as described below (lines 146-148):
The low density polyethylene (LDPE) pellets with 0.93 g/cm3 and a melt flow index (MFI) of 116 °C was used as a matrix and was purchased from B.O. packaging industry.
- SEM. Please provide the metal used for the coating.
Answer: In line 187 a new sentence was added:
and the samples were coated with gold.
- 1. The size distribution and the standard deviation must be given.
Answer: You are right, we improved the resolution and incorporated the histograms in order to study the distribution of the CaO into the LDPE matrix. The following paragraph was incorporated:
The pure CaO nanoparticles in the LDPE matrix were well distributed in the whole LDPE matrix, although with zones with some degree of agglomeration reaching the majority of agglomerations in higher frequency until 1-1.2 µm (histograms, Figures f and h). The nanoparticle dispersion was improved when nanoparticles modified with oleic acid (Mod-CaO) were added into the LDPE, the agglomerations decreased reaching high frequency until 0.5-0.6 µm (Figures 4g and i) confirming that their modification with oleic acid is an efficient route for improving filler dispersion in the matrix [11]. Therefore, the nanoparticle modification could increase the interaction between the modified nanoparticle surface and the polymer.
- A) Line 194-203. This discussion it is not clear, it is quite mixeitTIrd. Please improve it. B) FTIR, the peaks corresponding to the carbonyl groups to the nanocomposite.
Answer: In the DRX the following paragraph was improved in lines 236-243:
The CaO nanoparticles presented the characteristic Bragg reflections at 34°, 37°, 54°, 64° and 67° corresponding to planes (111), (200), (220), (311), and (222), respectively, related to the cubic structure of calcium oxide [24, 42,43]. Other peaks at 29°, 47°and 49° corresponding to planes (104), (018), and (214) are related to the calcite phase [24], which reveals how the fresh CaO easily reacts with carbon dioxide [44,45]. Furthermore, the presence of a peak at 18° corresponding to plane (0001) is consistent with the standard structure of portlandite [42], and it is due to the fact that CaO is very hydrophilic and reacts on contact with moisture in the atmosphere to produce small amounts of calcium hydroxide Ca(OH)2 [45].
- B) The IR is only from the nanoparticle
- 2. The SEM in the present form do not have much information an additional EDX to evaluate the aggregation of the CaO nanoparticles must be added. A cross section of the nanocomposite also could give an interesting information.
Answer: You have reason, we improve the resolution and incorporated the histograms in order to study the distribution of the CaO into the LDPE matrix. The next paragraph was incorporated
- Add a figure with the thermograms.
Answer: The DSC profiles was incorporated in to the new manuscript.
11.Mechanical properties. Provide the diagram of the mechanical properties. I suggest the addition of DMTA analysis to improve the discussion on the effect of the different nanoparticles on the nanocomposite.
Answer: You are right. The diagram of the mechanical properties was incorporated in to the new manuscript. The DMTA could improve the discussion, in particular the behavior at low temperatures, but we do not have access to this analysis. The following paragraph was incorporated in lines 400-408 in the new manuscript:
Torrano H.C et al. reported the dynamic-mechanical properties at 30-100 ºC when micrometric CaO nanoparticles were incorporated into the HDPE. The rigidity of the HDPE/CaO increased with the amount of particles at all temperature ranges in comparison with the neat HDPE. The increase of the storage modulus of the HDPE/CaO was higher at 30 ºC [35]. A similar behavior has been found by Kontuo et al. [55] for LDPE/SiO2 nanocomposites, where the storage and loss modulus increased with the amount of the nanoparticles compared to neat LDPE, particularly at low temperatures (-150 to -50 ºC).This increase may be due to the fact that macromolecular chains at the interface are restricted by the surface of the filler, greatly limiting molecular motion. An increase in the nanoparticle content enlarges the interfacial area and results in an increased interphase volume.
- Table 2. Please comment why Ee decreases in both cases at 5% but increases again at 10%.
Answer: The Reviewer is right pointing out that when CaO nanoparticles were incorporated into the polyethylene the Ee showed a decrease at both incorporations although at 10% the Ee is higher than at 5%, meaning a non-linear tendency. The reason for this behavior may be due to different processes competing at the same time, such as amount of particles, degree of agglomeration, and polymer structure.
The following paragraph in lines 392-4398 was incorporated:
The elongation at break of LDPE nanocomposites shows lower values than the pure polymer, although with a non lineal behavior, particularly with Mod-CaO particles of 25 nm diameter. In general, the elongation at break depended on the amount of filler and its dispersion in the polymer matrix, although other variables such as degree of crystallinity can further affect this property [53]. The tendency in our results can be associated with a competition between these variables that are different in the composites at 5 and 10 wt%. For instance, Sun et al. concluded that the use of nanometric particles makes difficult an explanation of the correct mechanism for elongation at break [51].
13.Figure 5. How many measurements have been carried out? Add the error bars for each measure. In addition, the evolution of the thermal and mechanical properties during this experiments should be evaluated.
Answer: For each sample three measurements were carried out. The new Figure 5 was incorporated with the error bars in the new manuscript. And the evolution of the mechanical properties is so difficult to measure due to the small sample analyzed.
The next sentence was incorporated into the experimental part in lines 219-221:
For each sample 3 measurement were carried out. The solutions were then centrifuged at 1800 rpm during 2 min to ensure the absence of any particulate matter in the solution. And each sample were measurement three times.
- Discussion, it is difficult to follow the discussion in the present format. The result and discussion must be together
Answer: You are right, but the Polymer Journal format ask first the results and after the discussion.

Round 2
Reviewer 1 Report
No more comments, the revised manuscript can be accepted for publication
Reviewer 2 Report
The authors have revised the manuscript well by reflecting my comments. Therefore, I recommend publication of this manuscript in current form.
Reviewer 3 Report
The article have been significantly improved by the incorporation of reviewers’ suggestion, I believe that the manuscript could be considered for its publication in the present form.